# Stability of Phenols, Antioxidant Capacity and Grain Yield of Six Rice Genotypes

**DOI:** 10.3390/plants12152787

**Published:** 2023-07-27

**Authors:** Juthathip Kunnam, Wanwipa Pinta, Ruttanachira Ruttanaprasert, Darika Bunphan, Thanasin Thabthimtho, Chorkaew Aninbon

**Affiliations:** 1Faculty of Agricultural Technology, King Mongkut’s Institute of Technology Ladkrabang, Bangkok 10520, Thailand; p_l_m_t-t@hotmail.com (J.K.); ktrung@kmitl.ac.th (T.T.); 2Faculty of Natural Resources, Rajamangala University of Technology Isan, Sakon Nakhon 47160, Thailand; wanwipanoi@gmail.com; 3Department of Plant Science, Textile and Design, Faculty of Agriculture and Technology, Rajamangala University of Technology Isan, Surin Campus, Surin 32000, Thailand; ruttanachira@gmail.com; 4Department of Agricultural Technology, Faculty of Technology Mahasarakham University, Maha Sarakham 44150, Thailand; darika.bu@msu.ac.th

**Keywords:** phenols, yield, rice, location, genotype, varieties

## Abstract

The environment is the main factor affecting variations in phytochemicals and antioxidant activity in rice. The objective of this study was to evaluate the stability of grain yield, phytochemicals and antioxidant capacity of six rice genotypes. Six rice genotypes were evaluated in a randomized complete block design with three replicates at three locations in Trat, Bangkok and Sakon Nakhon provinces in July–October 2019. Data on grain yield, yield components, total phenolic content, ferulic acid and antioxidant capacity were recorded. Grain yield was highest for crops grown in Bangkok, whereas antioxidant activity was highest for crops grown in Bangkok and Sakon Nakhon. Hom Nang Nual 1 and Mali Nil Boran had the highest grain yield. Riceberry had the highest grain yield in Trat; it also had high levels of total phenolic compounds, ferulic acid and antioxidant activity. Mali Nil Boran, Mali Nil Surin and Riceberry had the most stable total phenolic content, ferulic acid and antioxidant activity, respectively. Information on the levels and variability of phytochemicals in rice enables the selection of genotypes with high and stabile phytochemicals for production and rice breeding.

## 1. Introduction

Rice is a staple food crop that feeds more than half of the world’s population, and most of the rice production is in Asia [1]. Rice contains mainly carbohydrates. It also contains small amounts of other important nutrients such as proteins and minerals [2]. Rice consumers are currently concerned with their health, and there is an increasing trend towards consuming high-quality rice with additional health benefits.

Rice also contains phenol groups such as phenolic acids and flavonoids with antioxidant properties [3]. This group of substances plays a role in anti-oxidation [4] and plants survive in unsuitable environments [5]. Phenols found in rice grains include ferulic acid, p-coumaric acids, gallic acid, vanillic acid, caffeic acid and syringic acid, with ferulic acid being the most common compared with other phenolic acids [6]. Ferulic content is highest in purple bran rice, followed by red bran and light brown bran, with values of 1452.83, 1245.20, and 949.31 µg/g, respectively [7].

These phytochemicals with antioxidant activity are beneficial to health. Colored rice has higher phytochemicals and antioxidant activity than white rice [8]. However, acceptance of colored rice is still low and most of the rice sold in the market is white rice. Thailand does not currently dominate the world market, and the competition for white rice is high. The price of paddy rice also fluctuates and rice farmers in Thailand are at risk, although there are many government policies to help them. The demand for colored rice in Thailand is increasing. The promotion of colored rice is a way of diversifying rice products and reducing the production of white rice, which has market problems. It is more suitable for developing functional food products. According to Martirosyan and Singharaj [9], functional foods serve more functions than ordinary foods as they have more health and medical benefits, especially in the prevention of non-contagious diseases (NCDs). NCDs, including heart disease, stroke, cancer, diabetes and chronic lung disease are the major causes of death in the world [10]. 

The increase in phytochemicals with antioxidant activity adds value to rice and is an important strategy for adding quality to rice products. Therefore, in addition to high yield, the new breeding goal is to improve phytochemical compounds in rice grains [11]. However, the environment is an important factor that affects rice yield and accumulation of important phytochemicals [12]. The effects of genotype, location, year and the three-way interaction revealed significant differences in yield and other agronomic characteristics of rice in Ethiopia [13] and Thailand [14]. Moreover, the environment had large effects on the total phenolic content and total flavonoid content of red rice [11].

A lot of information on yield stability is available in the literature, as yield is most important for rice improvement. Grain yield is greatly affected by variations in climatic factors [15]. Genotypes with the highest grain yields were not stable [15]. The most stable genotypes and genotypes with adaptation to specific environments were identified. High yielding rice genotypes with adaptation to mid-hill environments were identified in Nepal [15]. In Ethiopia, rice genotypes with specific adaptations to six environments were identified [13] and in Thailand, rice varieties adapted to upland areas were identified [14]. For the stability of phytochemical responses to environments, most studies were conducted on fragrant rice [16,17]; however, information on phytochemicals and antioxidant activity is scarce, especially in Thailand. 

The environment was the main source of variation in total phenolic content, total flavonoid content and 2,2-azinobis-(3-ethylbenzothiazoline-6-sulfonic acid) (ABTS) radical scavenging capacity, accounting for more than 60% of the total variance [11]. According to Yamuangmorn et al. [18], the rice variety with high anthocyanin grown in the lowlands had higher grain yield compared with rice grown in the highlands; however, this variety grown in the lowlands had lower anthocyanin compared with plants grown in the highlands. For rice varieties with high antioxidant capacity, grain yield and 1,1-diphenyl-2-picrylhydrazyl (DPPH) activity were not significantly different between plants grown at the two elevations.

Selection of rice genotypes with good adaptation to a wide range of environments and specific environments is important for rice production and rice breeding. The varieties must be evaluated under a wide range of environments. Varieties showing consistent performance for a particular trait across environments are considered stable, while varieties showing good adaptation to specific environments are considered non-stable [19]. 

Several models are available for the evaluation of crop stability. Of these models, GGE biplot [13,20]) and AMMI [13] have been used. The model proposed by Eberhart and Russell [19] has been popularly used for analyzing stability using regression techniques. According to Eberhart and Russell [19], the genotype with a high mean yield, a regression coefficient (bi) that is close to one, and a deviation from regression that is nearly zero, is stable.

A better understanding of the effects of the environment and the interaction between genotype and environment on the stability of phenolic compounds, ferulic acid and yield allows breeders and producers to select suitable genotypes with general and specific adaptation to environments. The objective of this study was to evaluate the stability of phenolic acids, ferulic acid, antioxidant capacity and grain yield of six rice varieties.

## 2. Results

### 2.1. Effects of Environment, Genotype and Their Interaction on Traits

Differences between environments were significant (*p* ≤ 0.05 and 0.01) for number of tillers, number of panicles, 1000-grain weight, grain yield, ferulic acid and antioxidant activity as determined by the DPPH method, but the differences between the number of grains per panicle and total phenolic content (Table 1) were not significant. Differences between genotypes were significant (*p* ≤ 0.05 and 0.01) for most traits except number of grains. Interactions between genotype and environment (G × E) were also significant (*p* ≤ 0.05 and 0.01) for most traits except number of tillers, 1000-grain weight and total phenolic content.

The environment contributed to variations in these traits, ranging from 3.21% to 68.38% in total phenolic content and number of panicles, respectively. The contribution of environment to variations in the number of tillers, 1000-grain weight, grain yield and ferulic were intermediate, being 43.10, 49.19, 44.98 and 35.54%, respectively, whereas the contribution of the environment to variations in the number of panicles was high (68.38%) and the contributions of the environment to variations in the number of grains, total phenolic content and antioxidant activity as determined by the DPPH method were low, being 29.54, 3.21 and 11.63%, respectively.

The contributions of variety to total variations in the number of tillers, number of panicles, number of grains, 1000-grain weight and grain yield were low to intermediate, ranging from 11.20 to 32.26%, whereas the contributions of variety to total variations in total phenolic content, ferulic acid and antioxidant activity as determined by the DPPH method were high, ranging from 56.96 to 87.05%. The contributions of the interaction between genotype and environment were low to intermediate, ranging from 1.18 to 35.65%. The contributions of G×E interaction to variations in the number of tillers, number of panicles, number of grains, total phenolic content, ferulic acid and antioxidant activity as determined by the DPPH method were low, ranging from 1.18 to 20.76%, whereas the contributions of G × E interaction to variations in the number of grains was intermediate (35.65%).

### 2.2. Yield and Yield Components

As interactions between genotypes and the environment were significant for most traits, individual analyses of three environments were reported (Table 2). The crops grown in Trat and Sakon Nakhon had higher numbers of tillers than the crops grown in Bangkok. KDML105 had the highest number of tillers (18.53 tillers) across the three locations. The crops grown in Trat and Sakon Nakhon also had higher numbers of panicles than crops grown in Bangkok. Hom Nang Nual 1 and Riceberry had the highest numbers of panicles per plant, with 12.37 and 12.79 panicles per plant, respectively.

Differences between locations were not significant for the number of grains per panicles, and rice varieties on average were not sufficiently different across locations. However, differences between rice genotypes were significant for crops grown in Sakon Nakhon, and crops in Lhueang Thong had the highest number of grains (215.58 grains per panicle). The crops grown in Bangkok had the highest 1000-grain weight (24.26 g), whereas Hom Nang Nual 1 and Lhueang Thong genotypes had the highest 1000-grain weight (24.39 and 24.54 g, respectively). Hom Nang Nual 1 genotype from Trat and Sakon Nakhon had the highest 1000-grain weight, whereas the Lhueang Thong genotype from Trat had the highest 1000-grain weight.

Crops grown in Bangkok had the highest grain yield of 3014.2 kg ha^−1^, whereas Hom Nang Nual 1 and Mali Nil Boran genotypes had the highest grain yields (2907.3 and 2953.2 kg ha^−1^, respectively) across locations (Table 3). Hom Nang Nual 1 and KDML 105 had the highest grain yields in Sakon Nakhon (3156.7 and 3209.1 kg ha^−1^, respectively), Mali Nil Boran had the highest grain yield in Bangkok (3809.0 kg ha^−1^) and Riceberry had the highest grain yield in Trat (3228.0 kg ha^−1^).

### 2.3. Total Phenolic Content (TPC), Ferulic Acid and Antioxidant Capacity

Locations were not significantly different in terms of total phenolic content, with a range of 29.24–32.01 mg/100 g seeds (Table 3). Mali Nil Surin and Riceberry had the highest total phenolic content across the three locations. Mali Nil Boran and Mali Nil Surin also had the highest total phenolic content at the three locations. Riceberry had higher total phenolic content than all white rice genotypes at the three locations.

Location had a significant effect on ferulic acid content in rice. The highest ferulic acid content (21.37 mg/100 g seeds) was found in crops grown in Bangkok. Rice genotypes were significantly different at all locations and across three locations. Mali Nil Surin and Riceberry had the highest ferulic acid content in all locations and across three locations. Mali Nil Boran also had high ferulic acid content in all locations and across three locations, and was significantly higher than in all white rice varieties.

The highest antioxidant activity values determined by the DPPH method were found in crops grown in Bangkok (34.35%) and Sakon Nakhon (35.27%). Mali Nil Surin had the highest antioxidant activity (53.08%) across the three locations; it also had the highest antioxidant activity in Trat (42.65%) and Sakon Nakhon (71.68%). Mali Nil Boran and Riceberry also had high antioxidant activity values, which were significantly higher than all white rice genotypes across locations and at all locations. Analysis of white rice and black rice showed that black rice had significantly higher phenolic content, ferulic content and antioxidant capacity compared with white rice (Table 4).

### 2.4. Stability Analysis

Stability analysis results for grain yield, total phenolic content, ferulic acid and antioxidant capacity of six rice genotypes are shown in Table 5. According to Eberhart and Russell [19], a stable genotype is indicated by a high mean value, regression coefficient value of 1 and the smallest deviation from regression. For grain yield, most rice genotypes had regression values greater than one; however, Mali Nil Surin had the lowest regression coefficient (b = −0.075 *). Although Mali Nil Surin was more stable than other rice genotypes across environments, its grain yield was below average. Riceberry and KDML105 had very high values of deviation from regression, indicating high fluctuations in grain yield across environments.

High ferulic acid contents were found in Mali Nil Boran, Mali Nil Surin and Riceberry (Table 5). However, among these genotypes, only Mali Nil Surin had a regression coefficient lower than 1 (b = 0.808). Other rice genotypes with a regression coefficient lower than 1 were Hom Nang Nual 1, Lhueang Thong and KDML105. However, these rice genotypes had lower than average ferulic acid contents.

Most rice genotypes had stable total phenolic content. However, Mali Nil Surin and Riceberry were rather sensitive to their environments (b = 2.15 * and 2.06 **, respectively), but had high phenolic contents. Mali Nil Boran had the most stable total phenolic content, as it had a regression coefficient below 1 (b = 0.52) and a low deviation from regression. This variety also had higher than average phenolic content.

Hom Nang Nual 1, Lhueang Thong and KDML105 had stable antioxidant capacity (DPPH) as they had low regression coefficients (b < 1) and low deviations from regression; however, they also had low antioxidant capacity. Mali Nil Boran and Mali Nil Surin had high antioxidant capacity but they were sensitive to environmental changes (b = 2.75 and 2.46, respectively).

### 2.5. Correlation between Grain Yield, Yield Components and Antioxidant Compounds

For the relationships between grain yield and yield components, the number of tillers and number of panicles were positively and significantly correlated (r = 0.6807 **); grain yield and grain number were also positively and significantly associated (r = 0.5401 **) (Table 6). The number of tillers and number of panicles were not significantly correlated with grain number and grain yield.

All agronomic traits were not significantly correlated with total phenolic content, ferulic acid and antioxidant activity determined by the DPPH method. However, total phenolic content, ferulic acid and antioxidant activity determined by the DPPH method were positively and significantly correlated with each other, with the correlation coefficients (r) ranging between 0.6316 ** and 0.7140 **.

## 3. Discussion

### 3.1. Effects of Genotype and Genotype–Environment Interactions on Traits

In this study, genotype contributed to large proportions of total variations in most agronomic traits, except number of grains; it also contributed to large proportions of total variations in total phenolic content, ferulic acid and antioxidant activity as determined by the DPPH method. It is interesting to note here that the variations due to genotype were rather high for total phenolic content, ferulic acid and the antioxidant activity determined by the DPPH method.

Genotypic variations in grain yield and yield components have been widely studied in rice. Previous studies indicated wide variations for these traits [21,22,23] and it is possible to select rice genotypes with good yield and agronomic traits. Genotypic variations in phenolic content, ferulic acid and antioxidant activity determined by the DPPH method have been reported in previous studies [24,25]. According to Muntana and Prasong [26], the antioxidant activities of rice bran extracts were in the following order, from high to low: red > black > white rice brans. Similarly, for a wide collection of rice germplasms, rice accessions displayed an increasing order of total phenolic content in the white rice, red rice and black rice [8].

The interaction between genotype and environment played an important role in the variations in the number of panicles, number of grains and grain yield, but the effects were not significant for the number of tillers and 1000-grain weight. Its effects on the number of grains was rather large (35.65%) and this component could affect grain yield. A study of upland rice found that the environment contributed to 59.90% of the total variation in grain yield [14]. Rice yield was also affected by the environment [13,27].

In this study, the interaction between genotype and environment had significant effects on ferulic acid and the antioxidant activity determined by the DPPH method, whereas the effect on total phenolic content was not significant. The results indicate that selection of rice varieties for high total phenolic content is easier than selection for ferulic acid and antioxidant activity as determined by the DPPH method. Harakotr et al. [28] found that nutraceutical lipid compounds of rice such as α-tocopherol, γ-oryzanol, octacosanol and squalene was mainly affected by genotypes. Guo et al. [29] found that environmental parameters such as high altitude increased phenolic content in tartary buckwheat. However, in this study, the effect of the environment on ferulic content was large, accounting for 35.54% of the total variation. The results suggest that suitable environments may be selected for production of rice with high ferulic acid.

### 3.2. Stability of Grain Yield, Phenols and Antioxidant Capacity

In this study, six rice genotypes, including three colored rice varieties and three white rice genotypes, were evaluated for grain yield, yield components, total phenolic content, ferulic acid and antioxidant activity determined by the DPPH method at three locations, which are important rice production areas in Thailand. The aim of this study was to find rice varieties suitable for production in each area and rice varieties with good adaptation for grain yield and antioxidant activity in different environments.

Rice yields in this study ranged from 2012.3 to 3809.0 kg ha^−1^; Bangkok had the highest yield (3014.2 kg ha^−1^), whereas Trat had the lowest yield (2458 kg ha^−1^). Bangkok had the highest grain yield because it was the most suitable area for rice production prior to urbanization. The lowest yield in Trat could be due to excessive rainfall, high humidity (Figure 1) and poor soil fertility [30].

Stability studies allow breeders to identify genotypes with stable performance for important traits such as grain yield and quality traits across environments, and several methods are available for stability studies. According to Eberhart and Russell [19], a stable and well-adapted genotype is determined by mean value, regression coefficient and deviation from regression. A genotype with a regression coefficient higher than one is considered unstable and highly sensitive to environments; therefore, it is specific to niche environments. A regression coefficient lower than one indicates that the genotype is relatively stable, with greater tolerance to environmental changes.

Mali Nil Boran had the highest grain yield but was not significantly different from those of Hom Nang Nual 1, Riceberry and KDML105, whereas Mali Nil Surin had the lowest grain yield. However, Mali Nil Surin, which had the lowest grain yield, had the most stable grain yield (b = −0.075). Mali Nil Boran, which had the highest grain yield, was most sensitive to environments as grain yield was greatly reduced under unfavorable environments.

In this study, the ideal variety with high yield and yield stability was not found, as the most stable variety had low yield, while the high-yielding genotypes were more sensitive. The sensitivity of high-yielding genotypes to environments were found in rice [31] and other crops such as peanuts [32], maize [33] and soybean [34]. The high-yielding genotypes require more suitable environments to achieve high yield. Although they had higher yield reduction under unfavorable environments, their yield potential contributes greatly to higher yield under unfavorable environments [35].

However, varieties with the highest grain yields in each environment were identified. For example, Hom Nang Nual 1 and KDML105 were the best varieties for Sakon Nakhon, Mali Nil Boran was specific to Bangkok, and Riceberry was specific to Trat.

Rice genotypes used in this study were separated into two groups based on pericarp colors: white and black. Mali Nil Boran, Mali Nil Surin and Riceberry with black pericarp had higher total phenolic content, ferulic content and antioxidant capacity than those with white pericarp (Table 4). The results of this study supports previous studies with similar findings. Phenolic compounds and DPPH radical scavenging activity in black rice were higher than in brown rice and white rice [36,37,38].

Mali Nil Boran was the most stable genotype for total phenolic and ferulic contents, as its regression coefficients were near one and its standard deviations from regression were low. However, its means in each location and across locations were lower than those for Mali Nil Surin and Riceberry.

For antioxidant capacity (DPPH), most regression coefficients were different from one, indicating instability of these genotypes. Although these genotypes did not have stable antioxidant activity, Mali Nil Surin showed the highest antioxidant activity across locations.

### 3.3. Correlation between Grain Yield, Yield Components and Antioxidant Compounds

Relationships between traits are useful for the selection of multiple traits for plant breeding. The association between tiller number and panicle number could be due to the fact that most tillers have panicles on top. The lack of association between most agronomic traits such as grain number and grain yield with tiller number and panicle number could be due to the ability of rice crops to compensate these characters.

However, grain yield and grain number were positively and significantly associated. Selection of rice genotypes with high number of grains per panicle would result in high grain yield.

Antioxidant activity in rice can be determined by several methods, including 1,1-diphenyl-2-picrylhydrazyl (DPPH), 2,2′-azino-bis-3-ethylbenzthiazoline-6-sulphonic acid (ABTS) [39] and ferric reducing antioxidant power (FRAP) [40,41]. According to Shao et al. [36], total phenolics and ferulic acid were major phytochemicals contributing to antioxidant activity in rice. Phenolic content [8,42] and ferulic acid [43] were positively and significantly associated with antioxidant activity as determined by the DPPH method. Our results supported previous findings. The lack of association between yield-related traits and antioxidant activity indicated that improvement of antioxidant activity in rice could not affect yield.

The environment did not have a significant effect on total phenolic content, and the trait was due largely to the genotype. This might mean that the trait is less complex and genes controlling key components of phenolic acids should be identified in future investigations. Evaluation of recombinant inbred lines of contrasting parents or other related populations under a wide range of environments might reveal genetic markers associated with this trait. Moreover, stability studies of other phenols, including flavonoids, anthocyanin and proanthocyanins, and other phytochemical substances in rice, including gamma-aminobutyric acid (GABA) and gamma oryzanol, should be conducted.

## 4. Materials and Methods

### 4.1. Plant Materials and Experimental Design

Six rice genotypes popularly grown in Thailand were used in this study. Three genotypes were of white rice, including Hom Nang Nual 1, Lhueang Thong and KDML105, and three genotypes were of black rice, including Mali Nil Boran, Mali Nil Surin and Riceberry. These genotypes were selected because of their differences in grain colors and antioxidant activities [24]. These genotypes were laid out in a randomized complete block design with three replicates at three locations, including Trat province (12°14′37.10″ N, 102°30′54.50″ E) in the East of Thailand, Bangkok province (13.7563° N, 100.5018° E) in the Central Plain of Thailand and Sakon Nakhon province (17.1664° N, 104.1486° E) in the Northeast of Thailand. Heights above sea level were 25, 28 and 172 m for Bangkok, Trat and Sakhon Nakhon provinces, respectively.

Rice was planted in these locations during the main season on the same planting date in July 2019, and agronomic practices were the same in all the locations. The soil was ploughed twice and puddled. Rice bunds to separate the experimental plots were not constructed because fertilization and water levels were the same for all plots. Seedlings (25 days old) were transplanted at a rate of 1 seedling per hill on 2 × 5 m experimental plots, with a spacing of 25 × 25 cm. Days to anthesis after transplanting in Bangkok, Trat and Sakon Nakhon province were 81, 85 and 85 days, respectively (Figure 1). The crop cycles were similar, with only 3-day differences, and flowering was uniform. Days-to-harvest were 121, 124 and 123 days after transplanting in Bangkok, Trat and Sakon Nakhon province, respectively (Figure 1).

Compound fertilizer (formula 5-15-15 of N-P-K) was applied 15 days after transplanting (DAT) at a rate of 156 kg ha^−1^ and urea (46-0-0) was applied at pre-heading stage at a rate of 62.5 kg ha^−1^. Insecticides and herbicides were applied as needed to control weeds and insects.

### 4.2. Yield and Yield Components

All rice varieties were harvested in October 2019 and grain yield data were recorded. The bordered plants were harvested from a 20 m^2^ area in each plot. Grain yield was measured at approximately 14% of grain moisture content and calculated as grain yield per hectare. Ten plants were randomly chosen from each plot and used to measure the number of tillers per plant, the number of panicles per plant, the number of grains per panicle and 1000-grain weight.

### 4.3. Total Phenolic Content (TPC)

The total phenolic content was determined based on the method described by Rodnuch et al. [44]. Paddy rice samples were dehulled and ground into fine powder. The ground samples (2 g of each sample) were combined with 10 mL of methanol and incubated for 24 h at room temperature and then centrifuged at 5000 rpm to separate the liquid layer from the sediment. The liquid extracts were filtered through No. 4 Whatman filter paper and stored at 4 °C for further analysis. The total phenolic content was analyzed using Folin-Ciocalteu’s assay and reported as gallic acid equivalent (mg GAE/100 g dry weight of rice grain).

### 4.4. Ferulic Acid

Ferulic acid was measured using high-performance liquid chromatography (HPLC) based on the method suggested by Sawetavong et al. [45], with minor modifications. Ground samples of dehulled rice (2 g of each sample) were used for ferulic acid analysis. The ground samples were loaded into flasks, and 4 mL of methanol was added to each sample. The samples were mixed in a vortex mixer for 1 min at room temperature, transferred to a 30 °C water bath for 3 min and then centrifuged for 5 min at 2500 rpm. The rice extracts were collected and filtered through a 0.45 µm pore-size syringe filter before injecting into the C16 HPLC column. The mobile phase was methanol: acetic acid (99.5:0.5 *v*/*v*). The flow rate was 1.2 mL/min and UV detector was set to 326 nm. The chromatograms of ferulic acid for the six rice genotypes in each location are presented in Appendix A.

### 4.5. Antioxidant Capacity (2,2-Diphenyl-1-Picrylhydrazyl, DPPH)

Antioxidant activity was determined according to the DPPH method suggested by Kaur et al. [46], with minor modifications. Rice extract samples (0.1 mL) were loaded onto a 2.9 mL freshly prepared solution containing 0.1 mmol/L methanolic solution of DPPH. A 1 mL methanol solution was used as a control. After the samples were mixed, the mixed samples were incubated in the dark at ambient temperature for 30 min. Absorbance was recorded at a wavelength of 517 nm using a spectrophotometer. The percentage inhibition activity was calculated as follows:Scavenging activity (%) = [(A control − A sample)/A control] × 100%
where A control is the absorbance of the control sample and A sample is the absorbance of the test sample measured at 517 nm.

### 4.6. Data Analysis

Analysis of variance was performed for all data based on the experimental design using SAS 9.4 (SAS Institute Inc., Cary, NC, USA). Means were separated by Duncan’s multiple range test (DMRT). Stability analysis based on genotypic means was used to determine the consistency of genotype performance across environments and the factors contributing to the variations in the traits. Regression analysis was performed based on the method suggested by Eberhart and Russell [19] using the R-language program. Correlation coefficients between the parameters under study were calculated based on data from plots in three locations.

Regression is commonly and widely used to estimate the effect of environment on genotypes. It is a useful tool for crop evaluation, although many new techniques are available. According to Voltas et al. [47], biplot technique facilitates visual evaluation of the winning genotypes and the winning locations and is useful for recommending crop varieties. Regression also provides similar information.

## 5. Conclusions

The aim of this study was to select rice genotypes suitable for three rice production areas in Thailand, including a central plain area (Bangkok), the East (Trat) and the Northeast (Sakon Nakhon). The best location for grain yield was Bangkok, and the best locations for antioxidant activity were Bangkok and Sakon Nakhon. Hom Nang Nual 1 and Mali Nil Boran had the highest grain yield across locations. Mali Nil Boran also had the highest grain yield in Bangkok. Riceberry had the highest grain yield in Trat; it also had high total phenolic content, ferulic acid and antioxidant activity. Hom Nang Nual 1 and KDML105 had the highest grain yield in Sakon Nakhon. Mali Nil Surin was the most stable for grain yield, but it had low grain yield. The varieties with high grain yield were sensitive to environments for grain yield. Mali Nil Boran, Mali Nil Surin and Riceberry were most stable for total phenolic content, ferulic acid and antioxidant activity, respectively. Selection of rice genotypes specific to environments for grain yield and antioxidant activity would increase rice productivity and rice quality.

## Figures and Tables

**Figure 1 plants-12-02787-f001:**
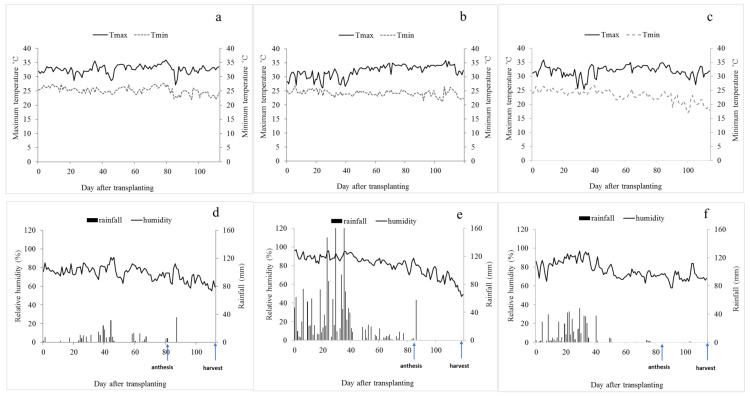
Maximum temperature (Tmax; °C), minimum temperature (Tmin; °C), relative humidity (%) and rainfall (mm) in Bangkok (**a**,**d**), Trat (**b**,**e**) and Sakon Nakhon (**c**,**f**) provinces during the growing season.

**Table 1 plants-12-02787-t001:** Mean squares of number of tillers per plant, number of panicles per plant, number of grains per panicle, 1000-grain weight, grain yield, ferulic acid and antioxidant activity of six rice genotypes as determined by the DPPH method evaluated across three locations in the main 2019 season.

Source of Variation	df	Number of Tillers/Plant	Number of Panicles/Plant	Number of Grains/Panicle	1000-Grain Weight	Grain Yield	Total Phenolic Content	Ferulic Acid	DPPH (%)
Environment	2	100.1 **	58.6 **	1911.5 ns	21.2 **	1,591,754 **	44.0 ns	100.0 **	321.6 *
		(43.10)	(68.38)	(29.54)	(49.19)	(44.98)	(3.21)	(35.54)	(11.63)
Rep. within site	6	5.0	6.2	313.4	1.6	74,698	37.9	8.3	113.6
		(2.15)	(7.23)	(4.84)	(3.71)	(2.11)	(2.77)	(2.95)	(4.11)
Genotype (G)	5	73.8 **	9.6 **	1177.0 ns	14.1 **	1,032,721 **	1199.1 **	160.3 **	1965.6 **
		(31.78)	(11.20)	(18.19)	(32.26)	(29.18)	(87.05)	(56.96)	(71.10)
G × E	10	48.2 ns	9.7 **	2307.4 **	3.0 ns	763,938 **	16.1 ns	9.3 *	301.6 **
		(20.76)	(11.32)	(35.65)	(6.96)	(21.59)	(1.18)	(3.30)	(10.91)
Error (G*Rep*E)	30	5.1	1.6	762.4	3.2	75,233	71.2	3.5	62.0
		(2.19)	(1.87)	(11.78)	(7.42)	(7.15)	(5.20)	(1.24)	(2.24)
Total	53								
C.V. (%)		17.68	10.99	15.81	7.86	10.25	27.73	10.14	24.31

ns = not significant; *, ** significant differences at *p* ≤ 0.05 and 0.001 level, respectively; numbers within the parentheses represent the percentages of mean squares or total mean squares.

**Table 2 plants-12-02787-t002:** Mean numbers of tillers per plant, panicles per plant, grains per panicle and 1000-grain weight (g) of six rice genotypes from individual analyses of three locations.

Genotype	Bangkok	Trat	Sakon Nakhon	Mean	Genotype	Bangkok	Trat	Sakon Nakhon	Mean
Number of tillers per plant					Number of panicles per plant				
Hom Nang Nual 1	9.05 ^cd1/^	14.07 ^bc^	13.17 ^b^	12.09 ^B2/^	Hom Nang Nual 1	8.95 ^b^	13.51 ^ab^	14.67 ^a^	12.37 ^A^
Lhueang Thong	11.29 ^ab^	12.06 ^c^	10.50 ^b^	11.28 ^B^	Lhueang Thong	10.38 ^b^	11.25 ^c^	9.42 ^b^	10.34 ^B^
KDML105	9.57 ^bc^	18.46 ^a^	27.55 ^a^	18.53 ^A^	KDML105	9.00 ^b^	12.12 ^bc^	14.75 ^a^	11.95 ^AB^
Mali Nil Boran	11.53 ^a^	14.20 ^b^	12.83 ^b^	12.85 ^B^	Mali Nil Boran	10.38 ^b^	13.20 ^abc^	12.83 ^a^	12.13 ^AB^
Mali Nil Surin	7.36 ^d^	14.84 ^b^	9.58 ^b^	10.59 ^B^	Mali Nil Surin	7.21 ^c^	14.47 ^a^	9.58 ^b^	10.42 ^B^
Riceberry	12.57 ^a^	14.63 ^b^	8.83 ^b^	12.01 ^B^	Riceberry	11.99 ^a^	14.23 ^ab^	12.17 ^ab^	12.79 ^A^
Mean	10.23 ^B^	14.71 ^A^	13.74 ^A^		Mean	9.65 ^B^	13.12 ^A^	12.23 ^A^	
F-test	**	**	**		F-test	**	*	**	
C.V. (%)	10.16	7.68	26.48		C.V. (%)	8.81	8.96	13.75	
Number of grains per panicle					1000-grain weight (g)				
Hom Nang Nual 1	178.33	155.53	153.33 ^cd^	162.40	Hom Nang Nual 1	24.37	23.51 ^a^	25.30 ^a^	24.39 ^A^
Lhueang Thong	189.67	162.60	215.58 ^a^	189.28	Lhueang Thong	26.26	23.37 ^a^	24.0 ^b^	24.54 ^A^
KDML105	162.20	174.47	180.67 ^bc^	172.45	KDML105	23.67	23.28 ^a^	22.57 ^c^	23.17 ^AB^
Mali Nil Boran	202.87	174.67	146.50 ^d^	174.68	Mali Nil Boran	23.17	21.27 ^b^	19.90 ^c^	21.45 ^C^
Mali Nil Surin	159.50	141.33	187.00 ^ab^	162.61	Mali Nil Surin	24.80	21.12 ^b^	21.13 ^d^	22.36 ^BC^
Riceberry	226.40	197.87	135.17 ^d^	186.48	Riceberry	23.22	20.87 ^b^	22.43 ^c^	22.17 ^BC^
Mean	186.49	167.75	169.71		Mean	24.26 ^A^	22.24 ^B^	22.56 ^B^	
F-test	ns	ns	**		F-test	ns	**	**	
C.V. (%)	21.50	11.16	10.68		C.V. (%)	12.17	3.87	2.68	

ns = not significant. *, ** Significant at *p* ≤ 0.05 and 0.01 probability levels. ^1/^ Means in the same column followed by a common letter are significantly different at *p* ≤ 0.05 by DMRT. ^2/^ Different capital letters indicate significant differences between environments and between genotypes at *p* ≤ 0.05 by DMRT.

**Table 3 plants-12-02787-t003:** Means of grain yield (kg ha^−1^), total phenolic content (mg/100 g seeds), ferulic acid (mg/100 g seeds) and antioxidant capacity (DPPH; %) of six rice genotypes from individual analyses of three locations.

Genotype	Bangkok	Trat	Sakon Nakhon	Mean
Grain yield (kg ha^−1^)				
Hom Nang Nual 1	3136.5 ^bc1/^	2428.7 ^bc^	3156.7 ^a^	2907.3 ^A2/^
Lhueang Thong	3063.0 ^bc^	2170.4 ^bc^	2616.3 ^b^	2616.5 ^B^
KDML105	2736.7 ^c^	2157.4 ^bc^	3209.1 ^a^	2701.1 ^AB^
Mali Nil Boran	3809.0 ^a^	2690.4 ^ab^	2360.2 ^bc^	2953.2 ^A^
Mali Nil Surin	2013.5 ^d^	2075.5 ^c^	2012.3 ^c^	2033.7 ^C^
Riceberry	3326.6 ^b^	3228.0 ^a^	1963.9 ^c^	2839.5 ^AB^
Mean	3014.2 ^A^	2458.4 ^B^	2553.1 ^B^	
F-test	**	**	**	
C.V. (%)	17.92	12.11	11.08	
Total phenolic content (mg/100 g seeds)				
Hom Nang Nual 1	21.27 ^cd^	20.64 ^bc^	24.06 ^b^	21.99 ^C^
Lhueang Thong	24.52 ^cd^	21.65 ^bc^	20.41 ^b^	22.19 ^C^
KDML105	17.16 ^d^	16.53 ^c^	20.64 ^b^	18.11 ^C^
Mali Nil Boran	30.26 ^bc^	35.60 ^ab^	33.43 ^b^	33.09 ^B^
Mali Nil Surin	44.29 ^a^	45.53 ^a^	50.64 ^a^	46.82 ^A^
Riceberry	37.93 ^ab^	39.25 ^a^	44.05 ^a^	40.41 ^A^
Mean	29.24	29.86	32.01	
F-test	**	*	*	
C.V. (%)	21.42	29.64	30.43	
Ferulic acid (mg/100 g seeds)				
Hom Nang Nual 1	16.25 ^b^	15.78 ^b^	14.13 ^c^	15.39 ^C^
Lhueang Thong	18.13 ^b^	15.35 ^b^	13.27 ^c^	15.58 ^C^
KDML105	15.02 ^b^	14.07 ^b^	12.44 ^c^	13.84 ^C^
Mali Nil Boran	25.28 ^a^	20.10 ^a^	20.49 ^ab^	21.95 ^AB^
Mali Nil Surin	26.28 ^a^	21.49 ^a^	23.79 ^a^	23.85 ^A^
Riceberry	27.27 ^a^	19.37 ^a^	17.75 ^b^	21.46 ^B^
Mean	21.37 ^A^	17.69 ^B^	16.98 ^B^	
F-test	**	**	**	
C.V. (%)	10.57	8.73	10.66	
Antioxidant activity (%)				
Hom Nang Nual 1	20.09 ^c^	19.55 ^c^	17.05 ^cd^	18.90 ^C^
Lhueang Thong	22.56 ^c^	20.52 ^c^	17.64 ^cd^	20.24 ^C^
KDML105	21.75 ^c^	20.55 ^c^	16.05 ^d^	19.45 ^C^
Mali Nil Boran	39.65 ^b^	27.64 ^bc^	53.49 ^ab^	40.26 ^B^
Mali Nil Surin	44.89 ^b^	42.65 ^a^	71.68 ^a^	53.08 ^A^
Riceberry	57.15 ^a^	34.30 ^ab^	35.73 ^bc^	42.39 ^B^
Mean	34.35 ^A^	27.53 ^B^	35.27 ^A^	
F-test	**	**	**	
C.V. (%)	17.23	22.09	30.26	

ns = not significant. *, ** Significant at *p* ≤ 0.05 and 0.01 levels. ^1/^ Means in the same column followed by a common letter are significantly different at *p* ≤ 0.05 by DMRT. ^2/^ Different capital letters indicate significant differences between environments and between genotypes at *p* ≤ 0.05 by DMRT.

**Table 4 plants-12-02787-t004:** Means of grain yield (kg ha^−1^), total phenolic content (mg/100 g seeds), ferulic acid (mg/100 g seeds) and antioxidant capacity (DPPH; %) of white rice and black rice from individual analyses of three locations.

Genotype	Bangkok	Trat	Sakon Nakhon	Mean
Grain yield (kg ha^−1^)				
White rice	2978.7	2664.6	2994.0 ^a1/^	2741.6
Black rice	3094.7	2252.2	2112.1 ^b^	2608.8
Mean	3014.2 ^A2/^	2458.4 ^B^	2553.1 ^B^	
Total phenolic content (mg/100 g seeds)				
White rice	20.98 ^b^	19.61 ^b^	21.70 ^b^	20.76 ^B^
Black rice	37.49 ^a^	40.12 ^a^	42.71 ^a^	40.10 ^A^
Mean	29.23	29.86	32.20	
Ferulic acid (mg/100 g seeds)				
White rice	16.46 ^b^	15.07 ^b^	13.27 ^b^	14.93 ^B^
Black rice	26.78 ^a^	20.32 ^a^	20.67 ^a^	22.42 ^A^
Mean	21.37 ^A^	17.69 ^B^	16.97 ^B^	
Antioxidant activity (%)				
White rice	21.46 ^b^	20.20 ^b^	16.91 ^b^	19.24 ^B^
Black rice	47.23 ^a^	34.86 ^a^	53.63 ^a^	45.24 ^A^
Mean	34.35 ^A^	27.53 ^B^	35.27 ^A^	

^1/^ Means in the same column followed by a common letter are significantly different at *p* ≤ 0.05 by DMRT. ^2/^ Different capital letters indicate significant differences between environments and between genotypes at *p* ≤ 0.05 by DMRT.

**Table 5 plants-12-02787-t005:** Stability analyses of grain yield, total phenolic content, ferulic acid and antioxidant capacity of six rice genotypes evaluated across three locations.

Genotype	Grain Yield (kg ha^−1^)	b	$s_{di}^{2}$	R^2^	Ferulic Acid (mg/100 g)	b	$s_{di}^{2}$	R^2^	Total Phenolic Content (mg/100 g)	b	$s_{di}^{2}$	R^2^	Antioxidant Activity (%)	b	$s_{di}^{2}$	R^2^
Hom Nang Nual 1	2907.3 ^A1/^	0.0854	189,776 **	0.9998	15.39 ^C^	0.366	−0.47	0.6030	21.99 ^C^	1.08	−21.01	0.8640	18.90 ^C^	−0.17	−19.32	0.1991
Lhueang Thong	2616.5 ^B^	1.403	25,226	0.8600	15.58 ^C^	0.992	−0.49	0.9194	22.19 ^C^	−1.15	−19.50	0.7305	20.24 ^C^	−0.12	−11.81	0.0426
KDML105	2701.1 ^AB^	0.384	503,790 **	0.7461	13.84 ^C^	0.48	−0.60	0.7516	18.11 ^C^	1.33	−20.78	0.8850	19.45 ^C^	−0.29	−8.65	0.1755
Mali Nil Boran	2953.2 ^A^	2.372	132,828 *	0.0853	21.95 ^AB^	1.192	−0.66	0.9514	33.09 ^B^	0.52	−8.79	0.9340	40.26 ^B^	2.75	41.12	0.8070
Mali Nil Surin	2033.7 ^C^	−0.075 *	−23,434	0.9464	23.85 ^A^	0.808	2.79	0.6303	46.82 ^A^	2.15 *	−21.89	0.9999	53.08 ^A^	2.46	280.45 **	0.4176
Riceberry	2839.5 ^AB^	1.062	930,297 **	0.9179	21.46 ^B^	2.16	−1.45	0.9991	40.41 ^A^	2.06 **	−21.89	0.9999	42.39 ^B^	1.37	237.10 **	0.2050
Mean	2675.2				18.68				30.44				32.39			

*, ** Significant at *p* < 0.05 and 0.01 probability levels; b = Regression coefficients; $s_{di}^{2}$ = Standard deviation of the regression coefficients; ^1/^ Different letters in the same column indicate significant differences at 95%; Mean comparison based on DMRT; R^2^ = regression coefficient.

**Table 6 plants-12-02787-t006:** Correlation coefficient (r) of number of tillers per plant, number of panicles per plant, number of grains per panicle, grain yield, total phenolic content (TPC), ferulic content and antioxidant capacity (DPPH) of six rice genotypes across three locations.

	No. Tiller	No. Panicles	No. Grain	Grain Yield	TPC	Ferulic Content
No. panicles	0.6807 **					
No. grain	0.0301	−0.1725				
Grain yield	0.2108	0.1295	0.5401 **			
TPC	−0.2071	−0.0260	−0.0409	−0.2436		
Ferulic	−0.3737	−0.2036	0.1943	0.0484	0.6316 **	
DPPH	−0.3226	−0.2375	0.0373	−0.2097	0.6420 **	0.7140 **

** Significant at *p* ≤ 0.01 probability levels.

## Data Availability

All data are available in the present work.

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
