# Peer review of "Stability of Phenols, Antioxidant Capacity and Grain Yield of Six Rice Genotypes"

_plants, 2023, doi:10.3390/plants12152787_

Round 1

Reviewer 1 Report

The objective of this study was to evaluate the stability of phenolic acids, ferulic acids, antioxidant capacity and grain yield of rice varieties (white and colored grains).

Six genotypes were evaluated in three locations and a unique year. It represent a very narrow a range of environments, that prevents encompassing environmental variability, especially variability between years. Authors must adequately justify the usefulness of this assessment in this limited environmental range. In addition, I detail other aspects that should be reviewed. In my opinion, the manuscript requires major revision before being published.

Introduction

L 50-53: How accepted is colored rice among consumers? Other functional foods, such as golden rice, had limited acceptance. Expand the idea of ​​the need for acceptance by consumers of functional foods.

L 62-63: ‘Therefore, the new breeding goal in addition to high yield is to improve phytochemical compounds in rice grain’ Please add citations.

L 65-67: ‘The effects of genotype, location, year, and the three-way interaction revealed the significant differences for yield and other agronomic characteristics of rice [24].’ Only one citation on this topic is not enough. Please, expand the citations and include updated reviews from rice-producing regions around the world, to give an adequate state of the art on the subject.

L 72-73: ‘The genotypes with highest grain yield were not stable [47]. The most stable genotypes and the genotypes with adaptation to specific environments were identified [24,47]’ For what environments? For what environmental restrictions? For which genotypes and grain quality? Expand. Also, explain carrefully the lack (if any) of GxE analysis for rice productivity and antioxidant capacity in Thailand. It is really important to justify the objective of the manuscritpt and justify publication despite the limited range of settings and a unique year evaluated.

L83-88: Please add citations.

L 99: Introduce the reader to the different phenolic compounds present in rice grains for both, white and colored, and the relevance of ferulic acid in particular. Also, briefly mention where the antioxidants are most concentrated in the grain (bran, endosperm, germ).

Results

L 105: Change the title ‘Analysis of variance’ for a more informative one bout the results obtained (not the statistical method used)

Tables 2 and 3: Order the genotypes for color (white and colored)

Total phenols had not significant effect for GxE, whereas ferulic acid and DPPH were significantly affected by GxE. To what do you attribute the differences? In addition, I suggest including the test for the differences between the set of white genotypes versus the colored one.

Table 4: Include the coefficient of determination (R2) to better inform the goodness of fit

Discussion

L 425-429: ‘Phenolic content [7, 40 ] and ferulic acid [41 ] ) were positively and significantly associated with antioxidant activity  determined by DPPH methods. Our results supported previous findings. The lack of association between yield related traits and antioxidant activity indicated that improvement of antioxidant activity in rice would  not affect yield.’ Since total phenols had not significant effect for GxE, do the authors consider this trait largely determined by the genotype? How could this be confirmed in future research? (expanding the range of environments for evaluation of the most contrasting genotypes, identifying markers for key genes in the synthesis of ferulic acid in the grain, etc). Please expand the discussion.

 M&M

Environment and Crop Phenology: Inform the reader about the climatic and soil differences among locations evaluated. Also, report the dates of heading, anthesis and physiological maturity for each variety x location combination (could add arrows in figure 1). Were the cycle lengths comparable to each other? Did they flower under similar conditions within the same locality? What level of stress (water, temperature) during grain filling did each genotype x location? This is very relevant to understand the level of grain yield achieved and the environmental conditions during the synthesis of phenolic compounds during grain filling.

L 459: ‘All bordered plants in each plot were harvested.’ Indicate the harvested area for each replicate, and the number of plants used for determine the number of panicles.

4.5. Define DPPH

4.6. Justify the method chosen to analyze genotype stability versus other methods mentioned in the Introduction.

Author Response

Dear Reviewer,

The manuscript is revised according to the comments and suggestions of the reviewers. Blue letters are indicated where the manuscript is changed. The new lines are also provided. Please see the attachment.

Best Regards,

Reviewer 2 Report

In further research, it is better to use wild, primitive rice varieties that have large amounts of phenols

No comment

Author Response

Dear Reviewer,

The manuscript is revised according to the comments and suggestions of the reviewers. Blue letters are indicated where the manuscript is changed. The new lines are also provided. 

Best regards,

Round 2

Reviewer 1 Report

The authors have improved the manuscript, but some points are still unclear.

Title: '....six rice' I suggest ....'six rice cultivars'

Lines 94-97: the phrase in blue is confusing, please clarify 

Line 97:   Define the acronym DPPH (since it is the mentioning for the first time)

Discussion section is too long and repetitive, focusing on GxE interactions based in only one experimental year. I suggest shortening the discussion by focusing on the objectives.

L 451-452: I don't agree with the statement.  There are information about genetic control of ferulic acid (and other antioxidants) in rice grain, especially for rice bran (and its cell wall). See the citations number 4 and 6 and refine the idea about the acknowledge gap. 

Revise the new phrases (in blue), especially in the title (six rice...?) and confusing phrases (for example, lines 94-96)

Author Response

Dear Reviewer,

The manuscript is revised according to the comments and suggestions of the reviewer. Blue letters are indicated where the manuscript is changed at the first time, and red letters are indicated where the manuscript is changed at the second time. The new lines are also provided.
